# Edaravone Attenuated Angiotensin II-Induced Atherosclerosis and Abdominal Aortic Aneurysms in Apolipoprotein E-Deficient Mice

**DOI:** 10.3390/biom12081117

**Published:** 2022-08-14

**Authors:** Haruhito A. Uchida, Tetsuharu Takatsuka, Yoshiko Hada, Ryoko Umebayashi, Hidemi Takeuchi, Kenichi Shikata, Venkateswaran Subramanian, Alan Daugherty, Jun Wada

**Affiliations:** 1Department of Chronic Kidney Disease and Cardiovascular Disease, Okayama University Faculty of Medicine, Dentistry and Pharmaceutical Sciences, Okayama 700-8558, Japan; 2Department of Nephrology, Rheumatology, Endocrinology and Metabolism, Okayama University Faculty of Medicine, Dentistry and Pharmaceutical Sciences, Okayama 700-8558, Japan; 3KYORIN Pharmaceutical Co., Ltd., Licensing Department, Tokyo 101-8311, Japan; 4Center for Innovative Clinical Medicine, Okayama University Hospital, Okayama 700-8558, Japan; 5Saha Cardiovascular Research Center, University of Kentucky, Lexington, KY 40506, USA; 6Department of Physiology, University of Kentucky, Lexington, KY 40506, USA

**Keywords:** edaravone, angiotensin II, abdominal aortic aneurysm, atherosclerosis

## Abstract

Background: The aim of the study was to define whether edaravone, a free-radical scavenger, influenced angiotensin II (AngII)-induced atherosclerosis and abdominal aortic aneurysms (AAAs) formation. Methods: Male apolipoprotein E-deficient mice (8–12 weeks old) were fed with a normal diet for 5 weeks. Either edaravone (10 mg/kg/day) or vehicle was injected intraperitoneally for 5 weeks. After 1 week of injections, mice were infused subcutaneously with either AngII (1000 ng/kg/min, *n* = 16–17 per group) or saline (*n* = 5 per group) by osmotic minipumps for 4 weeks. Results: AngII increased systolic blood pressure equivalently in mice administered with either edaravone or saline. Edaravone had no effect on plasma total cholesterol concentrations and body weights. AngII infusion significantly increased ex vivo maximal diameters of abdominal aortas and en face atherosclerosis but was significantly attenuated by edaravone administration. Edaravone also reduced the incidence of AngII-induced AAAs. In addition, edaravone diminished AngII-induced aortic MMP-2 activation. Quantitative RT-PCR revealed that edaravone ameliorated mRNA abundance of aortic MCP-1 and IL-1β. Immunostaining demonstrated that edaravone attenuated oxidative stress and macrophage accumulation in the aorta. Furthermore, edaravone administration suppressed thioglycolate-induced mice peritoneal macrophages (MPMs) accumulation and mRNA abundance of MCP-1 in MPMs in male apolipoprotein E-deficient mice. In vitro, edaravone reduced LPS-induced mRNA abundance of MCP-1 in MPMs. Conclusions: Edaravone attenuated AngII-induced AAAs and atherosclerosis in male apolipoprotein E-deficient mice via anti-oxidative action and anti-inflammatory effect.

## 1. Introduction

Abdominal aortic aneurysm (AAA) in humans is a common and potentially life-threatening condition [1]. AAA is located in the aortas, mostly distal to the renal arteries. The catastrophic consequence of AAA is rupture that usually has a fatal outcome. Approximately half of the patients with ruptured AAA die before administration to the hospital, while 30% and 50% die in the hospital [2,3]. Surgical repair is recommended when AAA expands more than 55 mm because of the increasing risk of rupture [4]. The several significant risks of the disease are identified as being over 65 years of age, smoking, family history of AAA, male sex, and chronic kidney disease [5,6]. The effective treatment for AAA is still limited to surgical operation [7,8,9,10]. Therefore, the development of new effective medical treatments to prevent AAA rupture is an important unmet medical need.

The roles of oxidative stress and antioxidants have been widely described in various previous reports. The involvement of highly reactive oxygen-derived free radicals (ROS) in the development and progression of various cardiovascular diseases, including left ventricular hypertrophy, coronary arterial disease, congestive heart failure, as well as aortic dilation, and aortic dissection, is documented [11,12,13]. Thus, a potential new therapeutic target of AAA is to diminish or manage oxidative stress.

Since it is difficult to define the mechanism of AAA in humans, experimental research relies on animal models. One of the most commonly used mouse models for AAA is chronic subcutaneous angiotensin II (AngII) infusion into hypercholesterolemic mice [14,15,16]. This model displays unique aspects compared to other models [17,18]. AngII infusion model presents AAA rupture and is associated with atherosclerosis and hypertension, similar to a human with AAA [19,20,21]. As AAA is a chronic inflammatory disease in which the aortic wall is expanded by the destruction of the vascular structure, the tissue of AAA is often accompanied by infiltration of monocytes and macrophages, differentiation and proliferation of smooth muscle, increased matrix metalloproteinases (MMPs) activity, and degradation of extracellular matrix, closely associated with AngII-induced oxidative stress and vascular inflammation [22,23].

Edaravone is a free-radical scavenger that exerts antioxidant activity [24]. In a clinical setting, edaravone is administered to patients in the acute phase of cerebral infarction. It is also given to patients with amyotrophic lateral sclerosis. Furthermore, the administration of edaravone may lead to a preferable effect on the clinical outcome of myocardial infarction [25]. Thus, the potential of edaravone for the treatment of cerebro- and cardiovascular disease is expected to lead to a preferable outcome through scavenging free radicals [26]. Several studies have reported its protective effect on cardiovascular diseases in different models, in part, via its anti-oxidative function [27,28,29,30]. However, it is unknown whether edaravone exerts a protective effect on AngII-induced AAA as well as atherosclerosis.

Given these previous findings, we hypothesized that edaravone would exert a beneficial effect against AAA formation and atherosclerosis. The present study aimed to investigate the effect of edaravone on AngII-induced vascular pathologies and its mechanism.

## 2. Materials and Methods

### 2.1. Mice and Study Protocol

Male 8–12-week-old apoE-deficient (apoE−/−) mice were purchased from The Jackson Laboratory (Bar Harbor, Maine, USA, Cat. #2052). All mice were maintained in a barrier facility, and ambient temperature ranged from 20 °C to 24 °C. Mice were fed with a normal diet and water ad libitum. Edaravone was kindly gifted from Mitsubishi Tanabe Pharmaceutical Company. Saline or edaravone (10 mg/kg/day) were started intraperitoneally one week prior to AngII infusion. AngII (1000 ng/kg/min, Bachem, Cat. #H-1705) or saline were infused via Alzet mini-osmotic pumps (Model 2004, Durect Corp) for 28 days. Mini-osmotic pumps were implanted subcutaneously on the right flank, as described previously [21,31]. For mRNA and protein (Western blotting and gelatin zymography) analyses, AngII or saline was infused into mice for 7 days. The experimental protocol was approved by the Ethics Review Committees for Animal Experimentation of Okayama University Graduate School of Medicine, Dentistry, and Pharmaceutical Sciences.

### 2.2. Blood Pressure Measurement

Systolic blood pressure and pulse rate were measured by a sphygmomanometer using a tail cuff system (BP-98A, Softron, Tokyo, Japan) as described previously [32]. In brief, unanesthetized mice were introduced into a small holder mounted on a thermostatically controlled warming plate and maintained at 37 °C during measurement.

### 2.3. Total Cholesterol Concentration Measurements

Blood samples were obtained by cardiac puncture while under anesthetic condition. Plasma total cholesterol concentrations were determined in individual plasma samples using commercially available enzymatic-based kits (Wako Chemicals, Cat. #439-17501) [33].

### 2.4. Quantification of Aneurysms and Atherosclerosis

After 28 days of AngII or saline infusion, mouse aortas were harvested for aneurysm and atherosclerosis quantification. Aortas were perfused with saline by left ventricular puncture and were fixed in formalin (10% wt./vol) overnight. Adventitial fat was removed, and the maximum external width of the suprarenal aorta was measured using computerized morphometry (Lumina Vision software, Mitani Corp. Available online: https://www.mitani-visual.jp/en/products/bio_imaging_analysis/lumina_vision/ accessed on 8 July 2022) as described previously [15,34]. Abdominal aneurysm was defined as a 50% increase compared with the saline-infused aorta. In the age-matched control mice, the mean suprarenal width was 0.80 mm; consequently, we defined AAA as >1.20 mm. Atherosclerosis was quantified on intima on aortic arches as % lesion area by en face method as described previously [34,35,36].

### 2.5. Histological Analysis

Mouse abdominal aortas were fixed in formalin, embedded in OCT compound, and were serially sectioned at 10 μm thickness. Sections were stained with Hematoxylin–Eosin. Verhoeff’s staining was used to examine elastin fiber integrity. Immunostaining was performed to examine macrophage accumulation using a CD68 antibody (Serotec, Kidlington, UK, Cat. #MCA1957). Reactivity of the antibodies with tissue antigens was detected using AEC and ImmPACT AEC HRP Substrate (Vector Laboratories) as described previously [32,34]. For fluorescent immunostaining, mouse aortas were frozen immediately after harvest, embedded in OCT, and serially sectioned at 10 μm thickness. Staining was performed to examine oxidative stress in aortic tissue using a 3-nitro tyrosine (3-NT) antibody (Abcam, Tokyo, Japan, Cat. #ab61392). Elastin breaks are defined as a disruption in the continuity of the lamina where both ends of the break are visible. Elastic fragmentation was quantified as elastin breaks per cross-section. The total number of breaks in the elastic lamella was manually quantified in each section by two observers blinded [37,38].

### 2.6. Gelatin Zymography

Either saline or edaravone injection (10 mg/kg/day) was started intraperitoneally one week prior to AngII infusion, and aortas were harvested one week after AngII infusion. Proteins were extracted from aortas without any distinguishable aneurysms. Samples were resolved under non-reducing conditions with polyacrylamide gels (7.5% wt./vol) containing gelatin (0.1% wt./vol) as the substrate for collagenase activity [15].

### 2.7. Isolation of Peritoneal Macrophages

Male 8–12-week-old apoE−/− mice were sedated, and peritoneal macrophages were harvested, as described previously [39]. Red blood cells were lysed using a solution of acetic acid. Cell numbers were calculated using a hemocytometer stained with trypan blue (Sigma, Saint Louis, MO, USA, Cat. #T8154). Next, saline or edaravone (10 mg/kg/day) was started intraperitoneally into male 8–12-week-old apoE−/− mice. After 3 days, mice were injected intraperitoneally with thioglycolate broth (Sigma, Cat. #B2551, 1 mL; 4% wt./vol). Seventy-two hours after thioglycolate injection, mice were sedated, and peritoneal macrophages were harvested. Red blood cells were lyzed using a solution of acetic acid. Cell numbers were calculated using a hemocytometer after being stained with trypan blue.

### 2.8. Peritoneal Macrophage Treatment

Harvested peritoneal macrophages were cultured with or without edaravone (1 μM) for 1 h, and cells were administered with or without LPS (Sigma, Cat. # L4391, 10 μg/mL) for 4 h.

### 2.9. Real-time Polymerase Chain Reaction

mRNAs were extracted from aortas or cell lysate of peritoneal macrophages using RNeasy Fibrous Tissue Mini kit (Qiagen, Germantown, MD, USA) or RNeasy Mini kit (Qiagen), respectively. Reverse transcription was performed using iScript cDNA synthesis kit (Bio Rad, Hercules, CA, USA). PCR reactions were performed with an ABI Step One Real-Time PCR System (Applied Biosystems, Foster City, CA, USA) using Fast SYBR Green Real-time PCR Mixture (Applied Biosystems). Primers for Ccl2/Mcp-1 and IL-1β were commercially available (Takara Bio Inc., San Jose, CA, USA). Each sample was normalized to values for 18s mRNA expression (ΔΔCT method).

### 2.10. Statistics

Data are presented as mean ± SEM. Statistical significance among groups was assessed by two-way ANOVA followed by the Holm–Sidak method, one-way analysis of variance with Student–Newman–Keuls post hoc, or one-way analysis of variance on Ranks with a Dunn’s post hoc, where appropriate, using SigmaPlot v14.0 (Systat Softoware. Inc., Richmond, CA, USA). Fisher’s exact test was used in the evaluation of the incidence of AAA. Values of *p* < 0.05 were considered statistically significant.

## 3. Results

Edaravone had no effect on body weight, blood pressure, and total cholesterol concentration. AngII increased systolic blood pressure in mice. Edaravone had no effect on body weight, systolic blood pressure, and total cholesterol concentration in apoE−/− mice with or without AngII infusion (Table 1).

Edaravone attenuated the formation of AngII-induced AAA and atherosclerosis to determine the effect of edaravone on the development of AngII-induced AAAs. Male apoE−/− mice administered with either saline or edaravone were infused with AngII for 28 days (AngII + saline; *n* = 17, AngII + edaravone; *n* = 16). Two mice died of aneurysm rupture in AngII + saline mice, and a mouse died of aneurysm rupture in AngII + edaravone mice. At the end of the study, ex vivo suprarenal aortic widths were measured (Figure 1A). In saline-infused mice, edaravone had no effect on ex vivo suprarenal aortic width (mean width of the abdominal aorta: 0.91 ± 0.23 mm in saline + saline mice, 0.93 ± 0.23 mm in saline + edaravone mice). In AngII + saline mice, AngII infusion significantly increased aortic width and formed AAAs (mean width of the abdominal aorta: 1.72 ± 0.13 mm, Figure 1A); however, edaravone administration attenuated enlargement of aortic width and decreased AAA formation (mean width of the abdominal aorta: 1.31 ± 0.13 mm, *p* = 0.032 vs. AngII + saline group, Figure 1A). The incidence of AAAs in AngII-infused mice without edaravone administration was 88%. Edaravone administration significantly decreased the incidence to 25% (*p* < 0.001, AngII + saline vs. AngII + edaravone Figure 1B). There was no significant difference in the aortic rupture rate between AngII + saline mice and AngII + edaravone mice (mortality rate: 12% and 6%, respectively, n.s.).

Next, the effect of edaravone on the development of AngII-induced atherosclerosis was investigated. In the saline mice, edaravone had no effect on atherosclerosis based on the en face lesion area (Figure 2). In AngII + saline mice, AngII infusion significantly increased en face lesion area compared with saline + saline mice (*p* < 0.001), however, edaravone administration attenuated AngII-induced atherosclerosis (*p* = 0.029).

Edaravone reduced medial disruption, macrophage accumulation, and oxidative stress to examine the histological characteristics of abdominal aortas. Verhoeff’s staining of cross sections of suprarenal aortas was performed. Pronounced disruptions of the medial layer were observed in AngII + saline mice. Disruptions induced by AngII infusion were markedly reduced in AngII + edaravone mice (Figure 3A,B). Since it is known that macrophage-mediated inflammation plays an important role in AngII-induced AAA formation, immunostaining to detect macrophage accumulation in aortas was performed. Immunostaining of CD68 revealed increased macrophage accumulation in the region of medial disruption in aortas from AngII + saline mice, whereas lesser macrophage accumulation in the medial layer was observed in AngII + edaravone-treated mice (Figure 3A).

Immunostaining of 3-NT was performed to investigate oxidative stress in aortas of the AngII-infused mice. AngII infusion enhanced 3-NT staining but was attenuated by co-administration of edaravone (Figure 3A).

Edaravone attenuated matrix metalloprotease activity in aorta gelatin zymography was performed to detect collagenase activities. After 7 days of AngII infusion, aortas without any distinguishable aneurysms were analyzed by gelatin zymography. Both pro-form MMP-2 (65 and 70 kDa) and active-form MMP-2 (58 kDa) were increased by AngII infusion and were attenuated by co-administration of edaravone (Figure 4A).

Edaravone attenuated the mRNA expression of inflammatory cytokines in the aortas gene expressions of inflammatory cytokines such as *Ccl2/Mcp-1* and *IL-1β* in the aorta increased in AngII + saline group. Edaravone administration attenuated the increase in mRNA expressions of these molecules (Figure 4B).

Edaravone attenuated thioglycolate-induced peritoneal macrophage accumulation and MCP-1 gene expression to evaluate the effects of edaravone on macrophage recruitment, and thioglycolate-induced peritonitis was applied as a model of inflammation [39]. Administration of edaravone significantly reduced the number of macrophages elicited to the peritoneal cavity after 72 h of thioglycolate injection (Figure 5A). In addition, mRNA abundance of *Ccl2/Mcp-1* in these macrophages increased by thioglycolate injection but was attenuated with co-administration of edaravone (Figure 5B).

Edaravone decreased LPS-induced MCP-1 mRNA abundance in macrophages. Finally, to investigate the effects of edaravone on inflammation-associated *Ccl2/Mcp-1* mRNA abundance, harvested peritoneal macrophages were stimulated with LPS and co-incubated with or without edaravone. LPS enhanced *Ccl2/Mcp-1* mRNA abundance that was reduced by co-incubation with edaravone (Figure 5C).

## 4. Discussion

This study demonstrated that edaravone attenuated AngII-induced AAA and atherosclerosis in apoE−/− mice, associated with a reduction in oxidative stress and inflammation in the aorta, a decrease in elastin degradation of the medial layer, and less accumulation of inflammatory cells in the vascular lesion. Furthermore, edaravone suppressed inflammation-elicited macrophages accumulation and incremented MCP-1 mRNA abundance in macrophages.

Emerging evidence has suggested that the development of AAAs in humans is related to vascular oxidative stress and inflammation [23,40,41,42,43]. Therefore, reducing oxidative stress and inflammation can be an efficient strategy for limiting AAA development [44]. Many studies have reported the similar significance of oxidative stress in the development of AngII-induced AAA [45,46,47,48]. In this experimental model, anti-oxidative compounds, such as vitamin E [49] and intermedin [50], demonstrated an inhibitory effect against AAA development. Since edaravone is a free-radical scavenger, it was expected that edaravone would suppress AAA formation through its anti-oxidative effect. Morimoto K et al. reported that edaravone inhibited both the formation and development of AAA in rats in elastase and extraluminal calcium chloride AAA models [51]. Although the experimental models they employed and edaravone doses were different from the present study, edaravone attenuated AAA formation associated with the reduction in oxidative stress in the aorta in all murine AAA models. Thus, our study confirms and extends the efficacy of edaravone on AAA and atherosclerosis development.

Atherosclerosis is the major underlying pathology of several cardiovascular diseases, including coronary artery disease, peripheral artery disease, and cerebrovascular disease. Currently, atherosclerosis is recognized as a chronic inflammatory disease [52,53,54,55]. Initiating processes of atherosclerosis include endothelial dysfunction, possibly caused by elevated low-density lipoprotein cholesterol, free radicals, and hypertension [52,53]. Renin-angiotensin system has consistently been shown to have a prominent role in atherogenesis in both humans and experimental animals. Chronic infusion of AngII promotes atherosclerosis in hypercholesterolemic mice independent of increases in blood pressure [14,15,56,57]. Thus, AngII infusion induces oxidative stress and vascular inflammation, which leads to endothelial dysfunction, promoting atherosclerosis. Promising drugs against AngII-induced atherosclerosis influence both pharmacological inhibitions of the renin-angiotensin system and the pro-oxidant properties. In the current study, edaravone administration attenuated atherosclerosis during AngII infusion into mice without any effect on systolic blood pressure. These results are consistent with an anti-oxidative property of edaravone against atherosclerosis.

The development of AAA and atherosclerosis has some common pathology, particularly the accumulation of inflammatory cells, especially macrophages, in human and experimental lesions [22,23,52,53]. MCP-1 (also referred to as CCL2) is a small inducible cytokine that belongs to the CC chemokine family, which recruits macrophages and several other inflammatory cells. Since MCP-1 is one of the key components of experimental AAA development [58,59], a strategy to reduce MCP-1 can provide an optimal effect to attenuate AAA development as well as atherogenesis. In the present study, AngII infusion enhanced macrophage accumulation in vascular lesions but was reduced by edaravone consistent with the drug reduced inflammation, which was associated with a decrease in MCP-1 and IL-1β mRNA abundance in the aorta. In addition, edaravone attenuated inflammation-elicited macrophage recruitment and LPS-enhanced mRNA abundance in macrophages, suggesting an anti-inflammatory effect of edaravone. Collectively, it is likely that edaravone directly decreases macrophage recruitment in the vascular site as well as expression of MCP-1 in both tissue and macrophages, reducing matrix degradation.

Several studies showed the involvement of the JAK2/STAT3 pathway in the development of AngII-induced AAA [60,61]. Namely, blocking the JAK2/STAT3 pathway attenuates experimental AAA development [62]. Since AngII induces oxidative stress, it is likely that edaravone protects AngII-induced AAA formation by its anti-oxidative effect through the inhibition of JAK2/STAT3 pathway activation. On the other hand, Chen H et al. reported the sharp contrast result that edaravone activates the JAK2/STAT3 pathway [63]. In this report, the acute myocardial infarction model was used in rats without the infusion of AngII. Therefore, the condition of oxidative stress in the tissue between these two models might be different. Target cells are different, myocardiocytes and vascular cells. In addition, myocardial infarction is a necrotic lesion due to ischemia; meanwhile, AAA is a dilated lesion. The phenotypes of these diseases are different. Finally, the dosage of edaravone and duration of infusion is different.

The strength of our study is that edaravone has already been launched and is already in clinical use, at first, for the treatment of patients with cerebral infarction, but recently amyotrophic lateral sclerosis was added as an indication [64]. Additionally, very recently, edaravone has gained attention as a potential drug to reduce specific side effects of various therapies for cancers [65]. Future clinical studies may warrant the expansion of indications for several diseases, including AAA and atherosclerosis. However, several limitations should be noted. Edaravone can be administrated only via drip infusion; in addition, it sometimes induces adverse effects, including acute kidney injury, nephrotic syndrome, liver dysfunction, fulminant hepatitis, thrombocytopenia, rhabdomyolysis, and anaphylactic shock. Thus, for clinical application, careful attention must be paid to administration methods and side effects.

## 5. Conclusions

In conclusion, edaravone attenuated AngII-induced atherosclerosis and AAA formation through anti-oxidative action and anti-inflammatory effects. Thus, edaravone can be one of the therapeutic agents for cardiovascular disease, especially atherosclerosis and AAA.

## Figures and Tables

**Figure 1 biomolecules-12-01117-f001:**
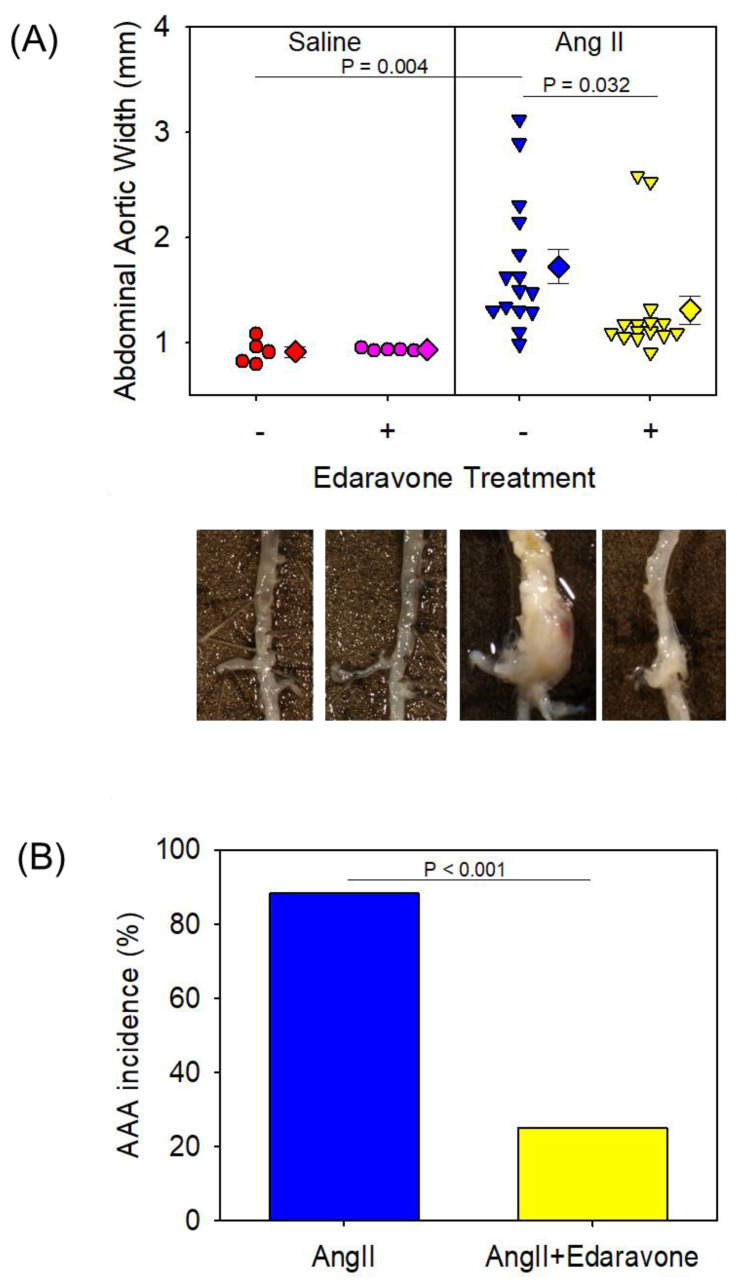
(**A**) Abdominal aortic width of each group. Representative images of the aorta in each group are shown. (**B**) Percent incidence of abdominal aortic aneurysm in AngII-treated groups. AngII: angiotensin II; AAA: abdominal aortic aneurysm.

**Figure 2 biomolecules-12-01117-f002:**
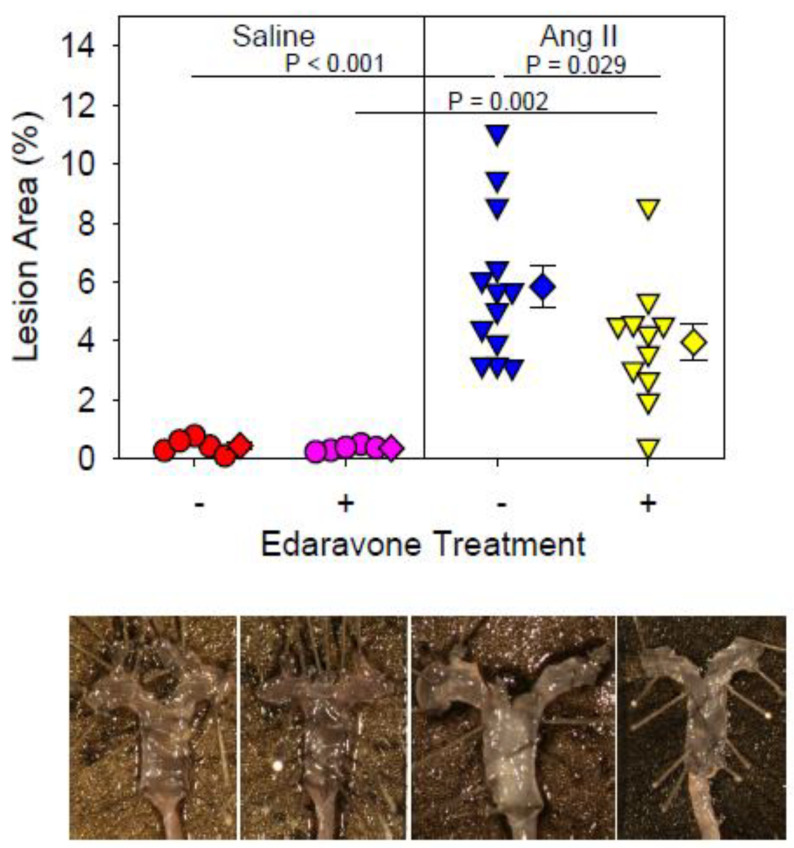
*En face* atherosclerotic lesion of each group. Representative images of *en face* atherosclerotic lesions in each group are shown. AngII; Angiotensin II.

**Figure 3 biomolecules-12-01117-f003:**
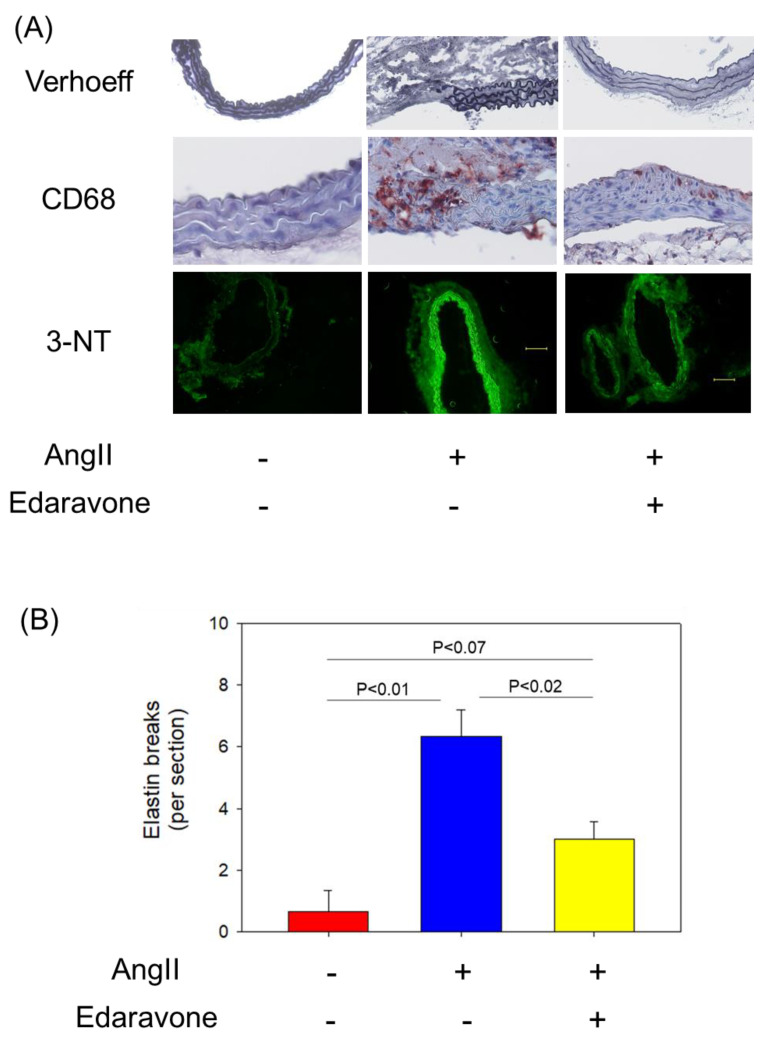
(**A**) Representative images of Verhoeff, CD68, and 3-NT staining of aortas in control, AngII + saline, and AngII + edaravone groups. (**B**) Elastin break of each group. Values are present as mean ± SEM (*n* = 3, each). AngII: angiotensin II; 3-NT: 3-nitrotyrosine.

**Figure 4 biomolecules-12-01117-f004:**
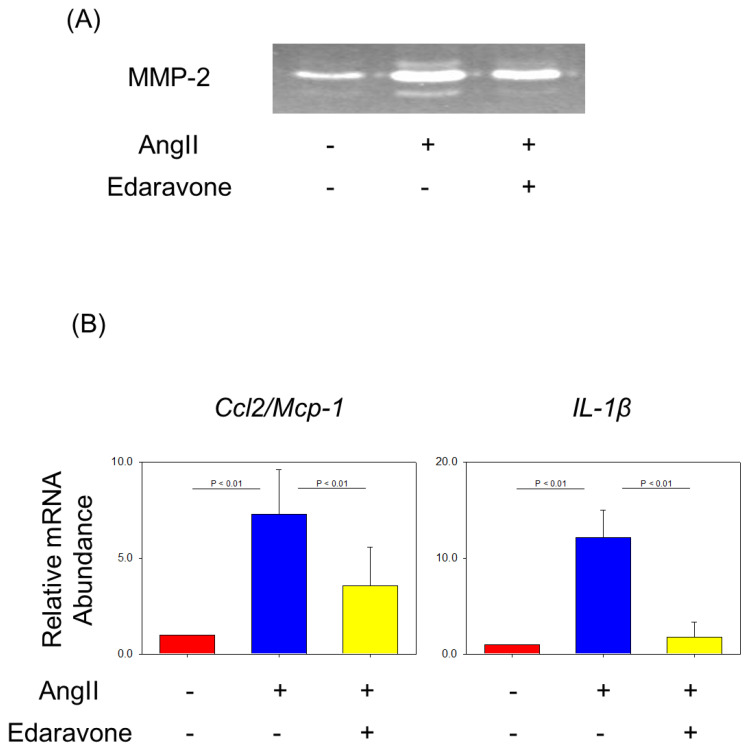
(**A**) Representative images of gelatin zymography. (**B**) Relative gene expression of *Ccl2/Mcp-1* and *IL-1β* of aortas in control, AngII + saline and AngII + edaravone group (*n* = 4 to 6). Values are present as mean ± SEM. AngII: angiotensin II.

**Figure 5 biomolecules-12-01117-f005:**
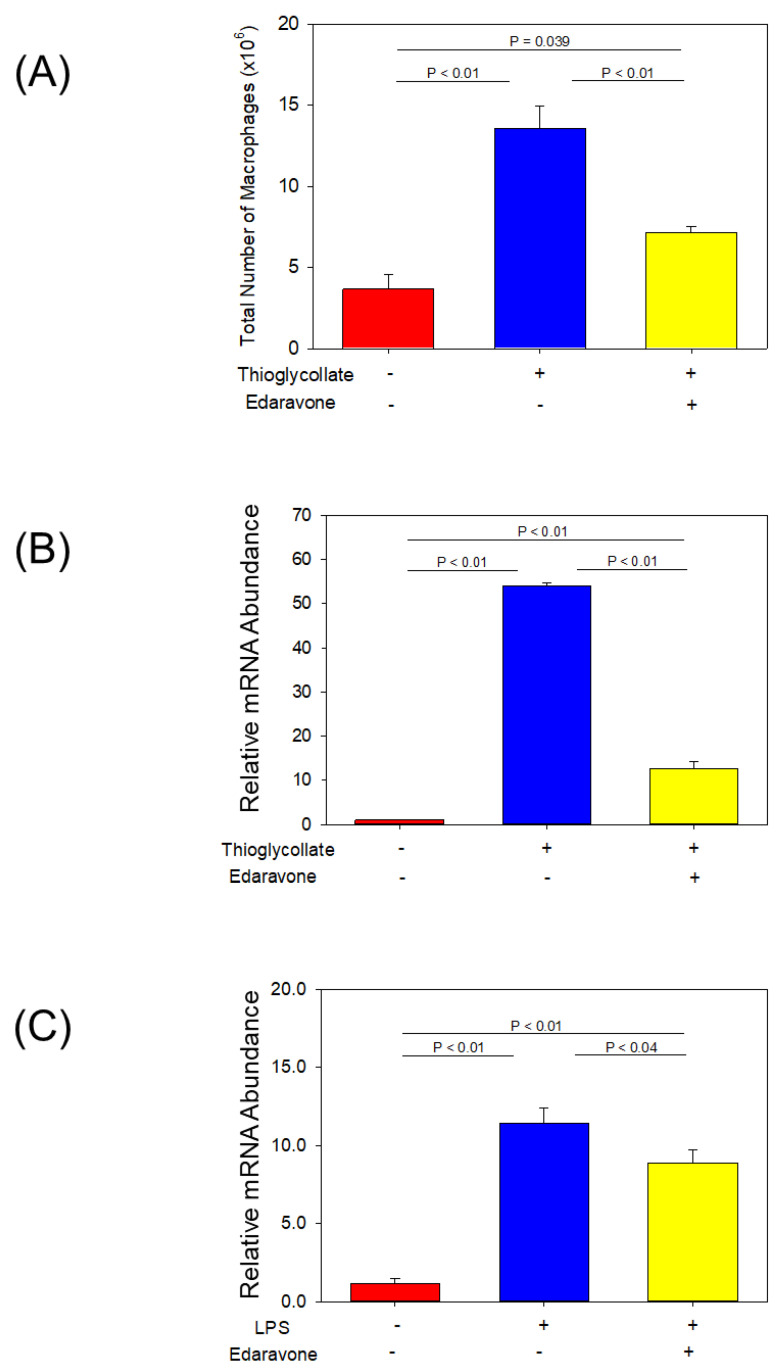
(**A**) Total number of macrophages (×106) in control, thioglycolate + saline and thioglycolate + edaravone group (*n* = 3). Values are present as mean ± SEM. (**B**) Relative gene expression of *Ccl2/Mcp-1* of macrophages in control, thioglycolate + vehicle and thioglycolate + edaravone group (*n* = 6, each). Values are present as mean ± SEM. (**C**) Relative gene expression of *Ccl2/Mcp-1* of macrophages treated with control, LPS + saline or LPS + edaravone for 4 h (*n* = 6, each). Values are present as mean ± SEM. LPS (10 μg/mL); lipopolysaccharide.

**Table 1 biomolecules-12-01117-t001:** Effects of edaravone administration on male apoE−/− mice infused with either saline or AngII.

	Saline		AngII	
	Saline	Edaravone	Saline	Edaravone
N	5	5	17	16
Body Weight (g)	28 ± 1	28 ± 0	28 ± 1	27 ± 1
SBP (mmHg)	97 ± 4	96 ± 4	122 ± 3 *	128 ± 3 *
T.Cho (mg/dl)	444 ± 38	476 ± 58	466 ± 27	464 ± 16

Body weight, systolic blood pressure, and serum cholesterol concentrations were measured at the end of the study. AngII; angiotensin II, SBP; systolic blood pressure, T.Cho; total cholesterol concentration. Values are represented as mean ± SEM. * denotes *p* < 0.05 when comparing AngII vs. saline by two-way ANOVA.

## Data Availability

The data sets generated during and/or analyzed during the current study are not publicly available as they have not been anonymized; however, they are available from the corresponding author on reasonable request.

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
