# Peer review of "Edaravone Attenuated Angiotensin II-Induced Atherosclerosis and Abdominal Aortic Aneurysms in Apolipoprotein E-Deficient Mice"

_biomolecules, 2022, doi:10.3390/biom12081117_

Round 1
Reviewer 1 Report
We congratulate the research team for a wonderful study.
Author Response
I am happy for your comments.
Reviewer 2 Report
The manuscript by Haruhito et al., submitted for publication, is an in-vitro and in-vivo study investigating Edaravone's therapeutic effect and mechanism on Angiotensin II-induced Atherosclerosis and Abdominal Aortic Aneurysms in Apolipoprotein E Deficient Mice. This is an exciting work with potential clinical implications, especially concerning the pathological structure of abdominal aortic tissue and inflammatory mediators. The study is well designed; the manuscript is well organized, well written, and relatively easy for the reader to follow. The reviewer would like to offer the following points for consideration by the authors towards the improvement of the manuscript:
1- How was the number of animals determined? Was there a power calculation?
2- What was the rationale for choosing the dose of Edaravone 10mg/kg/day?
3- The authors need to highlight the limitations of the current study.
Author Response
Comments and Suggestions for Authors
The manuscript by Haruhito et al., submitted for publication, is an in-vitro and in-vivo study investigating Edaravone's therapeutic effect and mechanism on Angiotensin II-induced Atherosclerosis and Abdominal Aortic Aneurysms in Apolipoprotein E Deficient Mice. This is an exciting work with potential clinical implications, especially concerning the pathological structure of abdominal aortic tissue and inflammatory mediators. The study is well designed; the manuscript is well organized, well written, and relatively easy for the reader to follow. The reviewer would like to offer the following points for consideration by the authors towards the improvement of the manuscript:
Thank the reviewer #2 so much for constructive comments to improve our manuscript. We here made point-by-point answers below.
1- How was the number of animals determined? Was there a power calculation?
Thank the reviewer #2 for important comments. To use essential minimal number of mice to find a significant effect, we first used 5 mice per group, then performed a power analysis. Seen in our previous papers, approximately 15-to-20 mice has been usually required for the experiment of AngII infusion model to examine AAA, as is in the current study.
2- What was the rationale for choosing the dose of Edaravone 10mg/kg/day?
Hang Xi et al. used 10 mg/kg i.p. infusion of Edaravone into apoE KO mice to investigate its effect on endothelial damage and earlier atherosclerosis (Atherosclerosis 191 (2007) 281–289). Okabe et al. used 10 mg/kg per day administration of Edaravone into apoE KO mice to examine its effect on high-fat induced atherosclerosis. (Circ J 2006; 70: 1216–1219). Both 2 papers finally found a significant effect of Edaravone on atherosclerosis.
As known well, chronic AngII infusion in hyperlipidemic mice induces AAA and atherosclerosis, so, we decided to use this dose of Edaravone in the current study.
3- The authors need to highlight the limitations of the current study.
Thank the reviewer #2 for constructive comments. I agree with you that this study has some limitation. First, Edaravone is widely used in a clinical setting as a treatment for the patient in the acute phase of cerebral infarction and amyotrophic lateral sclerosis. (page 2 line 73) Also, edaravone may exert a preferable effect on clinical outcome of myocardial infarction (page 2 line 75). Very useful.
However, several limitations should be noted. Edaravone can be administrated only via drip infusion, in addition, sometimes induces adverse effect including acute kidney injury, nephrotic syndrome, liver dysfunction, fulminant hepatitis, thrombocytopenia, rhabdomyolysis, anaphylactic shock, and so on. Thus, for clinical application, careful attention must be paid to administration methods and side effects. These are added in the text (page 13 line 345)
This manuscript is a resubmission of an earlier submission. The following is a list of the peer review reports and author responses from that submission.
Round 1
Reviewer 1 Report
The present study by Haruhito A. Uchida et al. is an investigation to define whether Edaravone participates in AAA formation. The authors use the model of AngII/ApoE-induced AAA formation in mice, and show that Edaravone attenuate AAA and atherosclerosis formation. Vascular CD68 positive macrophage infiltration and oxidative stress is also inhibited by Edaravone.
The study is overall well conducted and is of certain clinical interest. However, some drawbacks exist, which should be addressed as mentioned below.
Major:
- Introduction of Edaravone and its connection to AAA is quite inadequate. Since Edaravone is a radical scavenger,introducing how is ROS involved in the occurrence and development of AAA would be necessary. What is the performance of Edavarone in clinical studies of cardiovascular disease?
- The results of animal experiments are either in four groups, three groups, or two groups, which is not clear enough to understand. Please unify them (it would best to show the results in four groups like aortic width).
- The data of animal experiments may not conform to the normal distribution, please give more details of statistical methods.
Minor:
- It has been reported that Edaravone activates the JAK2/STAT3 pathway (Free radical research, 54(5), 351–359.), but the activation of the STAT3 pathway promotes AAA (International journal of cardiology, 312, 100–106. ). How do the authors explain this contradiction? Please discuss.
- Please show representative pictures of aortic aneurysm and atherosclerosis lesion.
- Please give statistics of elastin fragmentation beside Verhoeff staining.
- Please revise English writing.
Reviewer 2 Report
The paper by Uchida et al. asked the question of whether edaravone influenced angiotensin II (AngII) -induced atherosclerosis and abdominal aortic aneurysms (AAAs) formation. This is an important question, however, the presented paper raises several unanswered questions:
1. I would like to ask the authors at which day they monitored these parameters? It would be better to check all these parameters with different time points. How do the authors explain the fact of lack of body weight change as Ang II is known to induce bodyweight loss?
2. How do the authors explain the fact of non-changed total cholesterol levels as it is known from the literature that ApoE knockout mice demonstrate high plasma triglyceride and cholesterol levels?
3. It will be good to provide pictures from en face determination of atherosclerotic lesions.
4. The pictures need to be taken with higher magnification and better resolutions. Especially in Verhoeff staining, some elastin breaks are visible in the AngII+edaravone group. I recommend quantifying the data. For 4-NT staining, I would recommend using a different fluorophore cause elastin has a really high autofluorescence signal. How negative control looks like? The authors mentioned “ AngII infusion enhanced 3-NT staining but was attenuated by 205 co-administration of edaravone” but there is no proof for that as there is no picture from control mice.
5. Did the authors check the activation of pathways regulating oxidative stress as e.g. Nrf2-dependent antioxidant response?
6. What about the loading control in zymography? It will be good to quantify the data. It will be better to use in situ gelatin zymography method to have more conclusive data. Which cells are responsible for the observed phenomenon?
7. Did the authors check the inflammatory markers in the serum or plasma? It will be more appropriate.
8. What was the phenotype of recruited macrophages?
9. I rather recommend checking the inflammatory markers on protein level.